# A kinetic model of the central carbon metabolism for acrylic acid production in *Escherichia coli*

**Alexandre Oliveira**[ID], **Joana Rodrigues**◎, **Eugénio Campos Ferreira**‡, **Lígia Rodrigues**[ID]‡,
**Oscar Dias**[ID]*◎

Centre of Biological Engineering, University of Minho, Braga, Portugal

◎ These authors contributed equally to this work.
‡ ECF and LR also contributed equally to this work.
* odias@deb.uminho.pt

**Data Availability Statement:** All model files are available at the following URL: https://cutt.ly/aaKineticModels. Models are online on the following URLs: https://www.ebi.ac.uk/biomodels/

## Abstract

Acrylic acid is a value-added chemical used in industry to produce diapers, coatings, paints, and adhesives, among many others. Due to its economic importance, there is currently a need for new and sustainable ways to synthesise it. Recently, the focus has been laid in the use of *Escherichia coli* to express the full bio-based pathway using 3-hydroxypropionate as an intermediary through three distinct pathways (glycerol, malonyl-CoA, and *β*-alanine). Hence, the goals of this work were to use COPASI software to assess which of the three pathways has a higher potential for industrial-scale production, from either glucose or glycerol, and identify potential targets to improve the biosynthetic pathways yields. When compared to the available literature, the models developed during this work successfully predict the production of 3-hydroxypropionate, using glycerol as carbon source in the glycerol pathway, and using glucose as a carbon source in the malonyl-CoA and *β*-alanine pathways. Finally, this work allowed to identify four potential over-expression targets (glycerol-3-phosphate dehydrogenase (G3pD), acetyl-CoA carboxylase (AccC), aspartate aminotransferase (AspAT), and aspartate carboxylase (AspC)) that should, theoretically, result in higher AA yields.

## Author summary

Acrylic acid is an economically important chemical compound due to its high market value. Nevertheless, the majority of acrylic acid consumed worldwide its produced from petroleum derivatives by a purely chemical process, which is not only expensive, but it also contributes towards environment deterioration. Hence, justifying the current need for sustainable novel production methods that allow higher profit margins. Ideally, to minimise production cost, the pathway should consist in the direct bio-based production from microbial feedstocks, such as *Escherichia coli*, but the current yields achieved are still too low to compete with conventional method. In this work, even though the glycerol pathway presented higher yields, we identified the malonyl-CoA route, when using

MODEL2010030001 https://www.ebi.ac.uk/biomodels/MODEL2010030002 https://www.ebi.ac.uk/biomodels/MODEL2010030003 https://www.ebi.ac.uk/biomodels/MODEL2010030004 https://www.ebi.ac.uk/biomodels/MODEL2010030005 https://www.ebi.ac.uk/biomodels/MODEL2010030006 https://www.ebi.ac.uk/biomodels/MODEL2010030008 https://www.ebi.ac.uk/biomodels/MODEL2010040001 https://www.ebi.ac.uk/biomodels/MODEL2010040002 https://www.ebi.ac.uk/biomodels/MODEL2010040003 https://www.ebi.ac.uk/biomodels/MODEL2010040005 https://www.ebi.ac.uk/biomodels/MODEL2010040006 https://www.ebi.ac.uk/biomodels/MODEL2010040007 https://www.ebi.ac.uk/biomodels/MODEL2010160002.

**Funding:** This study was supported by the Portuguese Foundation for Science and Technology(FCT) under the scope of the strategic funding of UIDB/04469/2020 unit. This article is also a result of the project 22231/01/SAICT/2016: "Biodata.pt – Infraestrutura Portuguesa de Dados Biológicos", by Lisboa Portugal Regional Operational Programme (Lisboa2020), under the PORTUGAL 2020 Partnership Agreement, through the European Regional Development Fund (ERDF). Alexandre Oliveira holds a doctoral fellowship (2020.10205.BD) provided by the FCT. Oscar Dias acknowledge FCT for the Assistant Research contract obtained under CEEC Individual 2018. The funders had no role in study design, data collection and analysis, decision to publish, or preparation of the manuscript.

**Competing interests:** The authors have declared that no competing interests exist.

glucose as carbon source, as having the most potential for industrial-scale production, since it is cheaper to implement. Furthermore, we also identified potential optimisation targets for all the tested pathways, that can help the bio-based method to compete with the conventional process.

## Introduction

Acrylic acid (AA) ($C_3H_4O_2$) is an important chemical compound that is one of the key components of superabsorbent polymers [1–3]. According to the Allied Market Research, in 2015, the global market for AA was valued at 12,500 million US dollars, and is expected to reach 19,500 million US dollars until 2022 [4]. Despite its economic importance, the vast majority of AA is still produced by the oxidation of propylene or propane in a purely chemical process [1,5,6]. Ergo, the principal method for AA production was found to be expensive, with a high energy demand, thus contributing to the planet's environment decay. Hence, the development of an innovative and sustainable biological production method has been attracting the attention of the scientific community [1,2,7]. In the last decade, several semi-biological methods have emerged and were optimised. These methods consist of the bio-based production of 3-hydroxypropionate (3-HP) and its subsequent chemical conversion to AA. Despite the substantial improvements obtained with these methods, this process involves a catalytic step that increases the production costs and environmental impact due to high energy demands [1–3,5]. Hence, the AA's production method should, ideally, be a bio-based direct route as, in theory, microbial feedstocks are less expensive, allowing a higher profit margin [1]. Moreover, a more sustainable bioprocess allows to decrease non-renewable resources dependence and $CO_2$ emissions.

Fortunately, in recent years, it has been proven that it is possible to use engineered *Escherichia coli* to convert glucose or glycerol into AA. Like in the semi-biological methods, the bioprocess is also divided into two main parts, the production of 3-HP and its subsequent conversion to AA. This part of the pathway, from 3-HP to AA, has not been extensively studied. So far, there are only three studies that successfully converted glucose or glycerol to AA in *E. coli* [1,2,7]. Nevertheless, the synthesis of 3-HP is well reported, and three distinct pathways for its production have been identified, namely the glycerol route, the malonyl-CoA route, and the β-alanine route. From these pathways, it is well established that the glycerol pathway is associated with the highest yields. However, one of the reactions of this route requires the supplementation of vitamin $B_{12}$ (Fig 1), which is an expensive practice at an industrial-scale production, hence a significant disadvantage of this route [8,9].

The bio-based method is currently considered a promising alternative to the conventional process as the production of 3-HP increased considerably in the last few years. Recently, studies reported productions of up to 8.10 g/L with the glycerol pathway [1], 3.60 g/L with the malonyl-CoA pathway [10], and 0.09 g/L with the β-alanine pathway [11]. However, the AA yields obtained by Tong et al. (2016) [2] (0.0377 g/L) and Chu et al. (2015) [1] (0.12 g/L) for the glycerol pathway, and Liu and Liu (2016) [7] (0.013 g/L) for the malonyl-CoA pathway, established that this process still needs to be optimised to compete with the currently used methods.

Taking these considerations into account, the main goals of this work are to identify the reactions of the known routes for AA production (glycerol, malonyl-CoA, and β-alanine pathways) and to determine which pathway have a higher potential for industrial-scale production. *E. coli*'s central carbon metabolism (CCM) kinetic models will be used to analyse the three pathways using either glucose or glycerol as carbon source. Finally, novel optimisation strategies to improve the AA yields of the three biosynthetic pathways will also be sought.

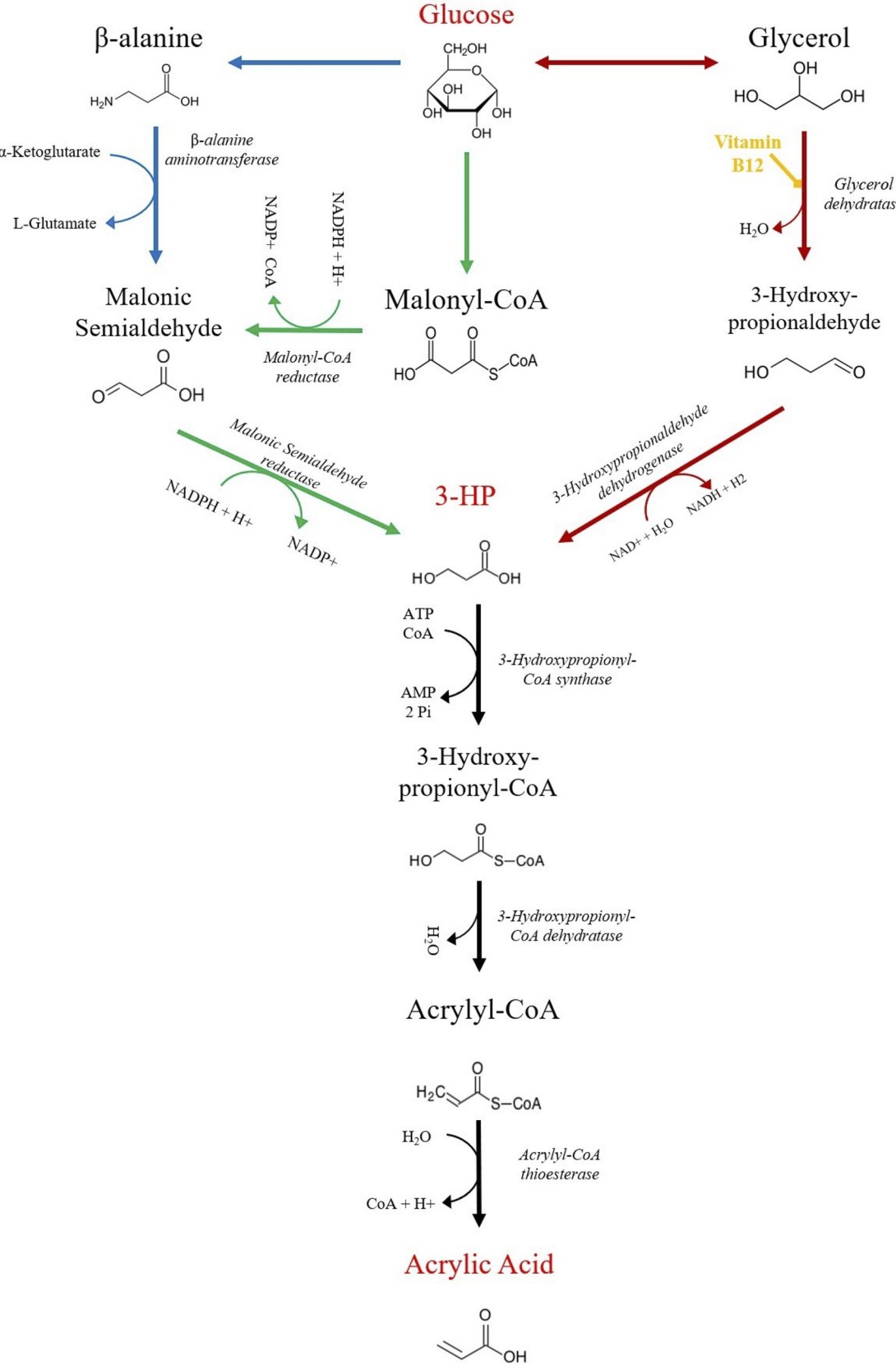

**Fig 1. Biosynthetic pathways for acrylic acid (AA) production from Glucose using 3-hydroxypropionate (3-HP) as an intermediary.** 3-HP can be produced from glucose through three distinct pathways: glycerol (red arrows), malonyl-CoA (green arrows), and *β*-alanine (blue arrows). Furthermore, *E. coli* can also direct glycerol towards the central carbon metabolism, allowing it to be used as a carbon source.

## Results and discussion

### Extended Central Carbon Metabolism (CCM_extended) model

Initially the original model of the CCM was extended to include the production of glycerol, malonyl-CoA, and β-alanine, from glucose (CCM_extended_Glc), resulting in a model with 87 reactions and 88 metabolites. This model is available at the Biomodels database with the identifier MODEL2010030001. Additionally, two more reactions were added to the CCM, which resulted in a new model with 89 reactions and 89 metabolites (Biomodels ID: MODEL2010160002), as the added reactions did not allow using glycerol as a carbon source. However, the latter model was not used to simulate the production of 3-HP and AA from glucose, as the it was not possible to determine all parameters of the GlyD reaction, which is responsible for the reversible conversion of glycerol to dihydroxyacetone. Method 1 only allowed the estimation of the $V_{max}$ parameter in the direction of dihydroxyacetone formation, due to the limitations of the stoichiometric model's flux balance analysis. Hence, only this direction was considered for the model, thus affecting the dynamic model behaviour when using glucose as a carbon source, as instead of contributing for glycerol biosynthesis, the reaction would deflect glycerol towards the CCM.

Although the original model was developed and validated for growth on glucose, the steady-state flux distribution (when using glycerol as carbon source) was compared with values determined experimentally [12,13]. This assessment unveiled a significantly different flux distribution between the dynamic model and experimental data (S1 Appendix, section 1. and Figs 1 and S1), which was considered when analysing the results under glycerol consumption. Addressing these differences would require determining the parameters for most reactions, which was not the goal of this work. Nevertheless, it would be a relevant topic to address in future work in order to improve the quality of the models.

### 3-Hydroxypropionate and acrylic acid producing models

The CCM_extended model was then used as a chassis, in which the three heterologous pathways were separately integrated, to simulate *in silico* production of 3-HP and AA from both carbon sources and to determine which is associated with higher yields. Twelve different dynamic models were generated, and the details of each model are presented in S1 Table. Moreover, during simulations, several issues arose, leading to variations in parameters before the analysis of the 3-HP and AA production. These variations are explained in detail in the S1 Appendix, section 1.2, and the results presented in S2–S5 Figs.

### Time course simulations

Regarding the production of 3-HP in the glycerol pathway, simulations with models set to use either glucose (Glu-Gly) or glycerol as carbon source (Gly-Gly), predicted, the production of 0.19 g/L (after three hours), and 8.30 g/L (after six-hours), respectively (Fig 2). Whereas, concerning the production of AA, the Glu-Gly and Gly-Gly models predicted 0.16 g/L and 6.71 g/L, respectively (Fig 3). From these results, glycerol seems to be associated with higher yields, which is in good agreement with the available literature [1]. Moreover, regarding the production of AA, the intracellular concentration of 3-HP showed that there is no accumulation

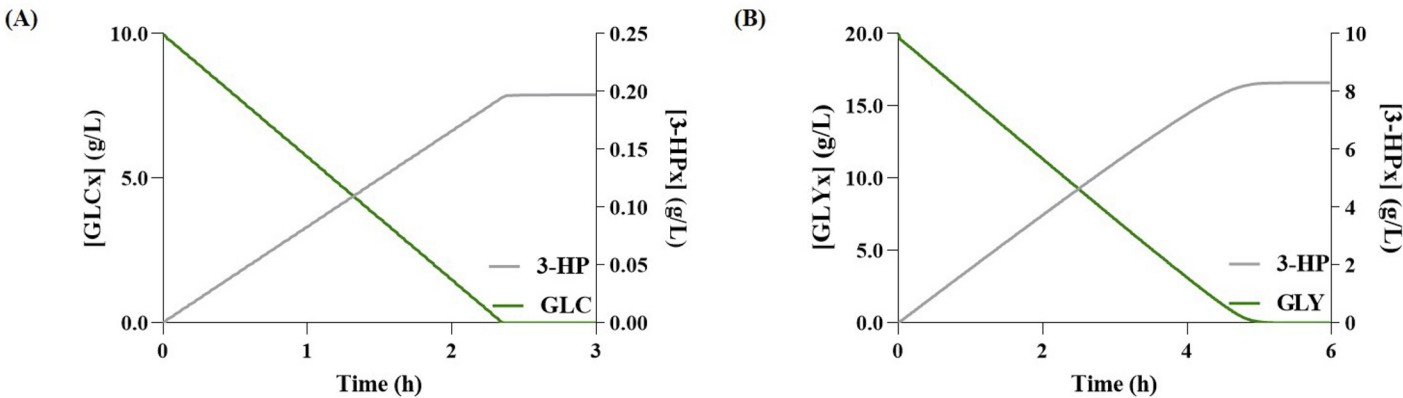

**Fig 2. Simulation results for 3-hydroxypropionate (3-HP) production via the glycerol pathway. (A)** Glucose (GLCx) consumption and variation of extracellular 3-HP (3-HPx) over time; **(B)** Glycerol (GLYx) consumption and variation of 3-HPx over time.

(Fig 3), meaning that most 3-HP is converted into AA. These results are most likely associated with the use of excessive enzyme concentration to calculate the $V_{max}$ for the heterologous pathway, which led to a state in which the main limiting factor in the synthesis of AA was the CCM's flux distribution. However, this is not the case *in vivo*, as the studies that tested the full

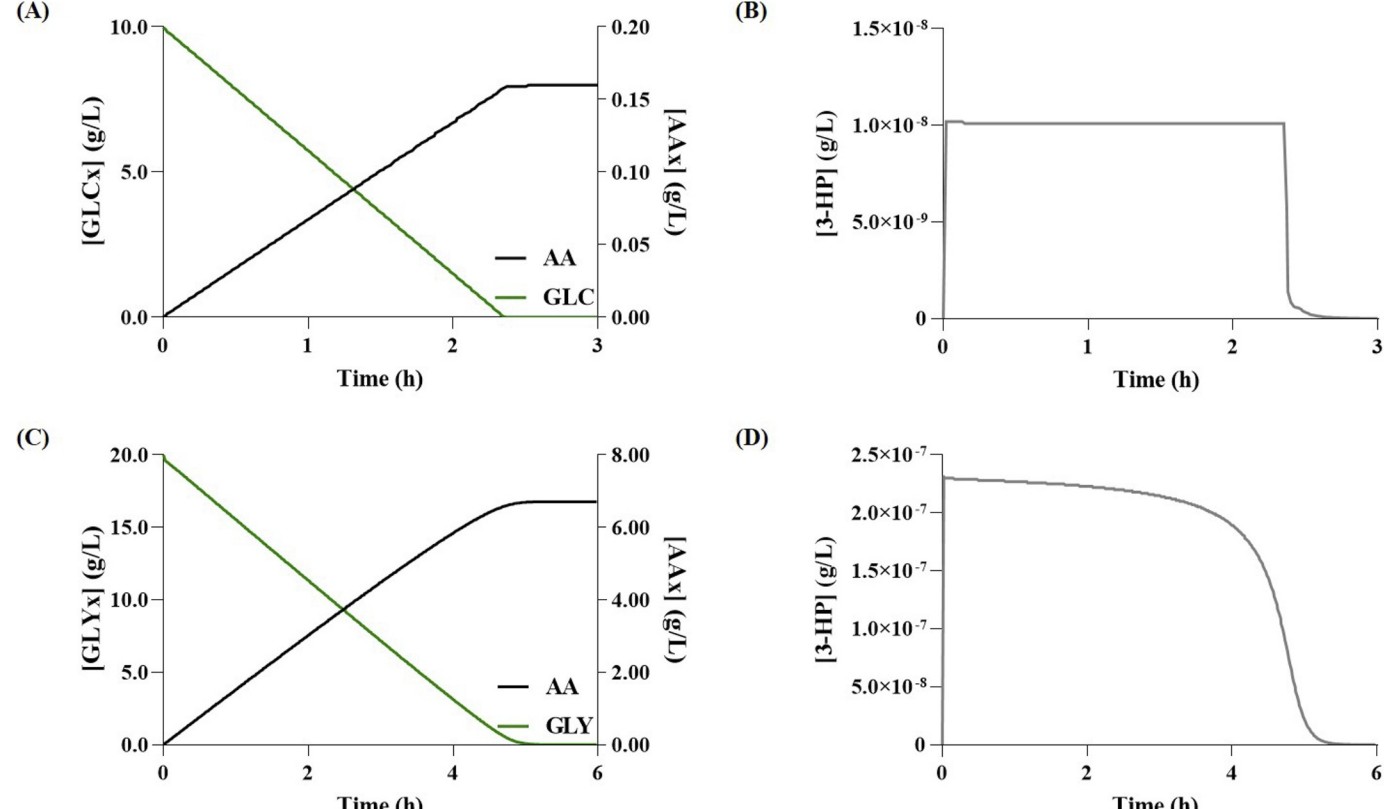

**Fig 3. Simulation results for acrylic acid (AA) production via the glycerol pathway. (A)** Glucose (GLC) consumption and variation of extracellular AA (AAx) over time; **(B)** Variation of 3-hydroxypropionate (3-HP) concentration over time when using glucose as carbon source; **(C)** Glycerol (GLYx) consumption and variation of extracellular AAx over time; **(D)** Variation of 3-HP concentration over time when using glycerol as carbon source.

bio-based pathway show that 3-HP and other intermediates indeed accumulate during this process [1,2].

When comparing the predictions of the glycerol pathway models (Table 1), it is possible to observe that the predicted 3-HP concentration, with the Gly-Gly model, is slightly different from literature reports. This difference slightly increases when increasing the initial concentration of carbon. For instance, for 40 g/L of glycerol, the predicted 3-HP production is about two times higher. Nevertheless, the model representing the 3-HP production from glycerol exhibits promising results.

The scenario with the Glu-Gly model is considerably distinct, as a ten-fold lower concentration of 3-HP was predicted. This difference might be associated with the production of glycerol, more specifically, in the flux through G3pD and G3pP, as according to Chu et al. (2015) [1] their strain is able to accumulate more glycerol (2.5 g/L) than this model is able to produce (0.34 g/L), for the same amount of glucose. Unfortunately, such study, which presented the highest AA concentration (0.12 g/L) reported thus far, did not disclose the amount of glucose used to obtain such production. Hence, it was not possible to directly compare the predicted titer from the Glu-Gly model.

When performing simulations using glucose concentrations of 10 g/L and 20 g/L, the model predicts the production of 0.16 g/L and 0.32 g/L of AA, respectively. These results show that the AA production predicted by our model would be in good agreement with the results reported by Chu et al. (2015) [1], if 10 g/L of glucose had been used for the carbon source. However, the model does not accumulate any intermediary compounds of the heterologous pathway; thus, most 3-HP is converted into AA, which does not correctly represent the *in vivo* results. Consequently, when assessing the results to the work of Tong et al. (2016) [2] that tested the production of AA in *E. coli*, the model fails to predict the AA yield accurately, as expected.

Regarding the malonyl-CoA pathway, the model set to use glucose as carbon source (Glu-Mcoa) predicted a titer of 1.99 g/L of 3-HP, while the malonyl-CoA model set to use glycerol as carbon source (Gly-Mcoa) predicted 1.99 g/L of 3-HP (Fig 4). Moreover, the Glu-Mcoa and Gly-Mcoa models predicted the production of 1.62 g/L and 0.17 g/L of AA, respectively (Fig 5). The behaviour analysis of the Gly-Mcoa model showed a considerable intracellular accumulation of dihydroxyacetone (Fig 4C). Hence, most carbon does not reach the CCM, and therefore these results should not be considered as it is not possible to determine the best carbon source to produce AA. Although literature reports suggest a consensus towards the use of glucose as a carbon source, it should be noted that no work using glycerol was found. Thus, glycerol should not be excluded as a promising alternative carbon source.

Regarding the Glu-Mcoa models, the 3-HP production predictions are very similar to those found in the literature (Table 2). However, the models failed to predict the production of AA, as Liu and Liu, (2016) [7] reported the accumulation of 3-HP, which was not replicated by the model (Fig 5).

**Table 1. Literature review on 3-hydroxypropionate (3-HP) and acrylic acid (AA) production yields by the glycerol pathway in metabolically engineered *Escherichia coli*, and comparison with the yields predicted by the dynamic models using the same initial carbon concentration.**

| Reference | End Product | Carbon Source | Initial Carbon Conc. (g/L) | Titer (g/L) | Predicted Titer (g/L) |
|---|---|---|---|---|---|
| Raj et al. (2009)[14] | 3-HP | Glycerol | 9.20 | 2.80 | 3.59 |
| Rathnasingh et al. (2009) [15] | 3-HP | Glycerol | 18.40 | 4.40 | 7.59 |
| Chu et al. (2015) [16] | 3-HP | Glycerol | 40.00 | 7.40 | 17.19 |
| Chu et al. (2015) [1] | 3-HP | Glycerol | 40.00 | 8.10 | 17.19 |
|  |  | Glucose | 21.50 | 3.90 | 0.42 |
| Tong et al. (2016) [2] | AA | Glycerol | 20.00 | 0.037 | 8.30 |

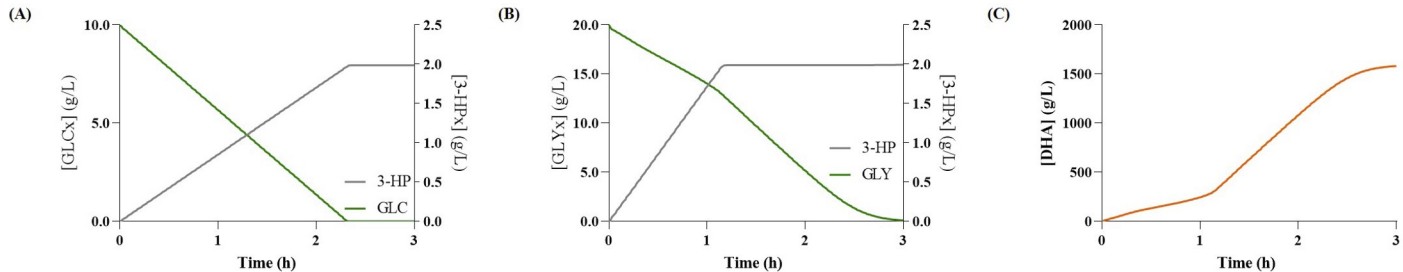

**Fig 4. Simulation results for 3-hydroxypropionate (3-HP) production via the malonyl-CoA pathway. (A)** Glucose (GLCx) consumption and variation of extracellular 3-HP (3-HPx) over time; **(B)** Glycerol (GLYx) consumption and variation of 3-HPx over time; **(C)** Variation of intracellular dihydroxyacetone (DHA) concentration over time when using glycerol.

Finally, regarding the $\beta$-alanine pathway, as shown in Figs 6 and 7, the $\beta$-alanine model set to use glycerol as carbon source (Gly-Ba) predicted the production of 0.041 g/L of 3-HP and 0.033 g/L of AA. The model set to use glucose as a carbon source (Glu-Ba) predicted the production of 0.033 g/L of 3-HP and 0.026 g/L of AA. Again, the models did not predict the accumulation of 3-HP that, although desirable, is not realistic. Although simulation results indicate a slight advantage towards using glycerol as a carbon source, there seems to be a consensus in literature towards using glucose as carbon source, as to the best of our knowledge, no studies

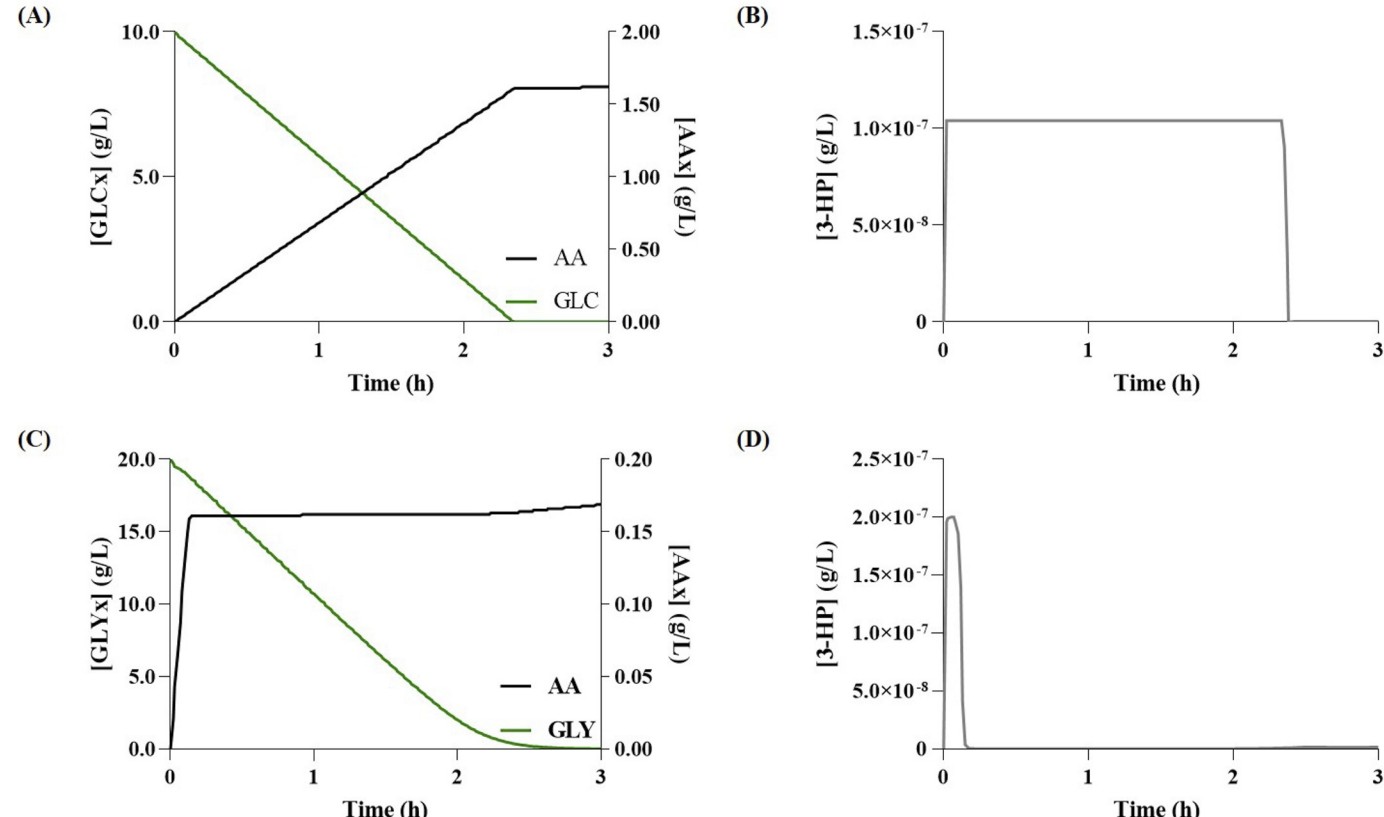

**Fig 5. Simulation results for acrylic acid (AA) production via the malonyl-CoA pathway. (A)** Glucose (GLC) consumption and variation of extracellular AA (AAx) over time; **(B)** Variation of 3-hydroxypropionate (3-HP) concentration over time when using glucose as carbon source; **(C)** Glycerol (GLYx) consumption and variation of extracellular AAx over time; **(D)** Variation of 3-HP concentration over time when using glycerol as carbon source.

**Table 2. Literature review on 3-hydroxypropionate (3-HP) and acrylic acid (AA) production yields by the malonyl-CoA pathway in metabolically engineered *Escherichia coli*, and comparison with the yields predicted by the dynamic models using the same initial carbon concentration.**

| Reference | End Product | Carbon Source | Initial Carbon Conc. (g/L) | Titer (g/L) | Predicted Titer (g/L) |
|---|---|---|---|---|---|
| Cheng et al. (2016) [17] | 3-HP | Glucose | 10.00 | 1.80 | 1.99 |
| Liu et al. (2016) [10] | 3-HP | Glucose | 20.00 | 3.60 | 3.96 |
| Liu and Liu (2016) [7] | AA | Glucose | 20.00 | 0.013 | 3.22 |

using the glycerol pathway have been reported. When compared to previous pathways, both carbon sources produced a considerably lower concentration of AA in this pathway.

The *β*-alanine pathway is the least studied, with very few reports, which might be associated with the fact that such studies reported significantly lower yields, when compared with the previous pathways [18]. Indeed, only one study was found concerning 3-HP synthesis in *E. coli* using batch cultures [11], while studies in which AA is produced through this route are yet to be published. The work of Ko et al. (2020) details the production of acrylic acid using *β*-alanine as intermediate; however, these authors found a novel pathway that bypassed the production of 3-HP. Thus, these results were not considered in the current study [19]. Nonetheless, the *β*-alanine model predictions showed promising 3-HP yields, as the projected concentration was close to the results obtained *in vivo* by Song et al. (2016) [11] (Table 3).

When comparing the three bio-based routes, the results suggest, as supported by literature [1,18], that the glycerol pathway leads to the highest yields, when combined with the use of glycerol as a carbon source. However, a relevant caveat must be recalled. This pathway includes a reaction that relies on the presence of vitamin $B_{12}$, which represents a significant economic disadvantage at an industrial-scale production [8,9]. Hence, to make this route economically viable, either the yield must be significantly improved to overcome the cost of the vitamin supplementation, or a cheaper path to produce $B_{12}$ must be found. Therefore, despite producing less AA, it seems to be beneficial to use the malonyl-CoA route, as it provided the second-highest yield and does not require vitamin supplementation [17,18,20]. Nevertheless, the pathway should still be optimised to obtain yields that could compete with the existing methods at an industrial-scale production.

## Optimisation strategies

Ideally, all models capable of producing AA should have been optimised. However, as mentioned before, the Gly-Mcoa model presented issues with dihydroxyacetone accumulation, not being further used in this work. Additionally, the Gly-Gly and Gly-Ba models proved to be unstable when performing the metabolic control analysis (MCA), preventing the flux control

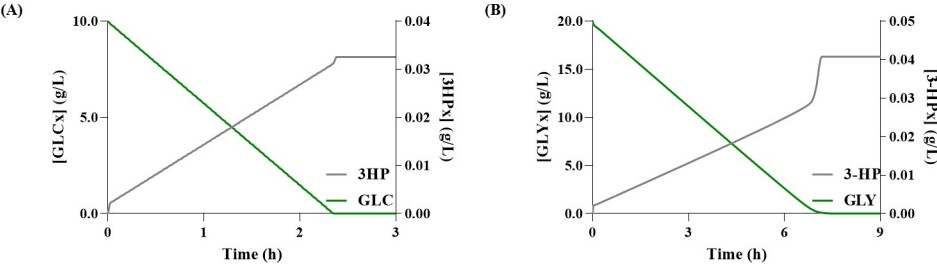

**Fig 6. Simulation results for 3-hydroxypropionate (3-HP) production via the *β*-alanine pathway. (A)** Glucose (GLCx) consumption and variation of extracellular 3-HP (3-HPx) over time; **(B)** Glycerol (GLYx) consumption and variation of 3-HPx over time.

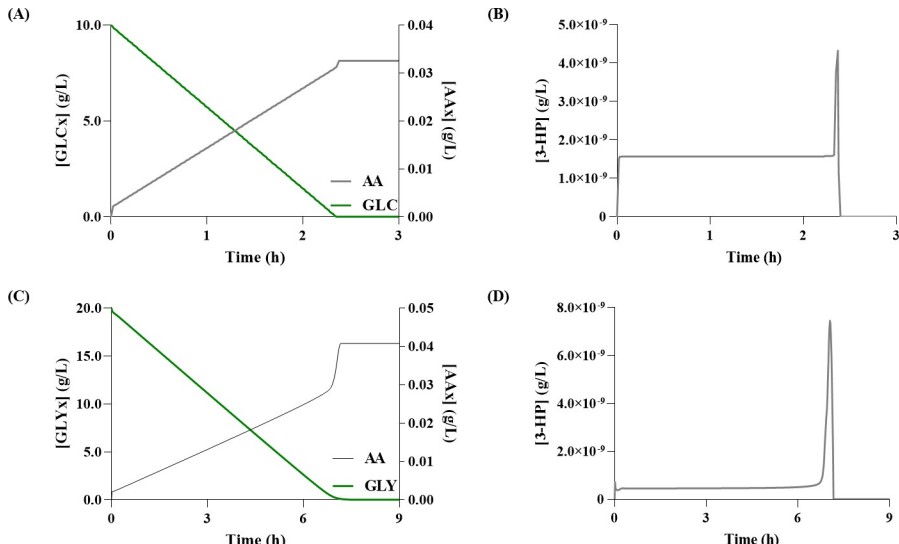

**Fig 7. Simulation results for acrylic acid (AA) production via the *β*-alanine pathway. (A)** Glucose (GLC) consumption and variation of extracellular AA (AAx) over time; **(B)** Variation of 3-hydroxypropionate (3-HP) concentration over time when using glucose as carbon source; **(C)** Glycerol (GLYx) consumption and variation of extracellular AAx over time; **(D)**—Variation of 3-HP concentration over time when using glycerol as carbon source.

coefficient (FCC) ascertainment, due to the lack of a steady-state. Henceforth, only the models designed to use glucose as a carbon source (Glu-Gly, Glu-Mcoa, Glu-Ba) were optimised.

Starting with the glycerol model, the MCA showed that the enzyme with the highest FCC, and thus a more significant influence on AA production, was the G3pD (Fig 8A), which is responsible for converting dihydroxyacetone phosphate into glycerol-3-phosphate. The reaction catalysed by this enzyme is a potential bottleneck in the pathway, and thus a target for overexpression. To the best of our knowledge, there are no evidences in literature regarding the use of this reaction as a target for optimisation, as this pathway is mainly used to produce 3-HP or AA from glycerol. Moreover, only Chu et al. (2015) [1] used glucose for AA production; however, they focused their optimisation strategies on limiting the toxicity of intermediary compounds.

Therefore, using COPASI's optimisation task, Mutant Glu-Gly 1 was created. This mutant included an *in silico* overexpression of the selected enzyme, in which the $V_{max}$ of the enzyme was set to 1.392 mM/s, representing a nearly 45-fold increase that resulted in the production of 3.11 g/L of AA (Fig 9A). A subsequent MCA was performed on Mutant Glu_Gly 1, aiming at further optimising the AA production yields. However, the model could not reach a steady-state; hence, the FCCs were not available, thus terminating the optimisation of this model.

In the malonyl-CoA pathway model, the FCCs identified one potential overexpression target, the AccC (Fig 8B), which is responsible for the conversion of acetyl-CoA to malonyl-CoA. Moreover, this is a well-established target for optimisation of the malonyl-CoA pathway [18]. The optimum $V_{max}$ for this enzyme was found to be 0.568 mM/s, corresponding to approximately a 2-fold overexpression. Mutant Glu-Mcoa 1 was able to produce a 3.11 g/L of AA,

**Table 3. Literature review on 3-hydroxypropionate (3-HP) production yields by the *β*-alanine pathway in metabolically engineered *Escherichia coli*, and comparison with the yields predicted by the dynamic models using the same initial carbon concentration.**

| Reference | End Product | Carbon Source | Initial Carbon Conc. (g/L) | Titer (g/L) | Predicted Titer (g/L) |
|---|---|---|---|---|---|
| Song et al. (2016) [11] | 3-HP | Glucose | 15.00 | 0.09 | 0.039 |

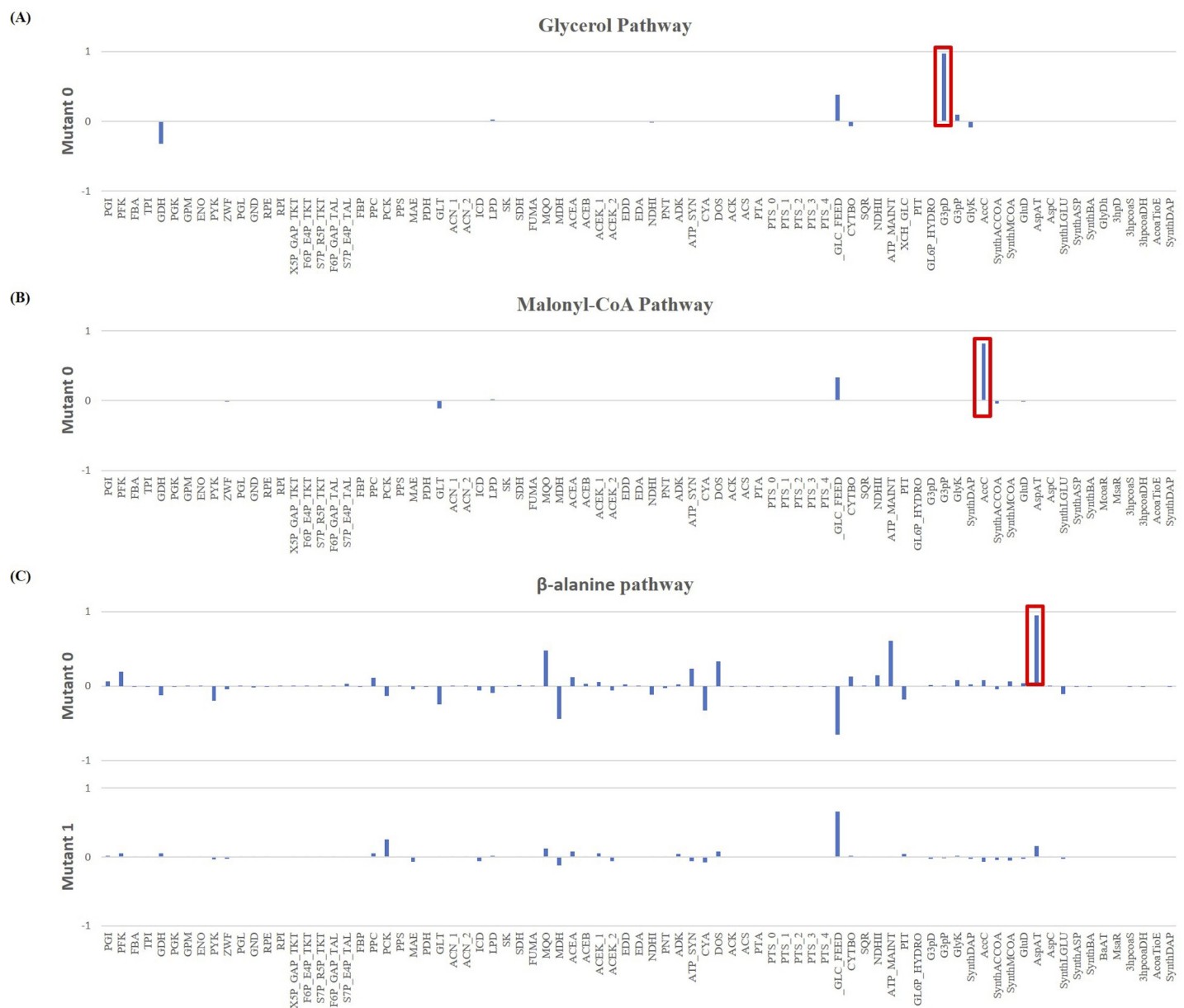

**Fig 8. Flux Control Coefficients (FCC) results for acrylic acid formation, where the reaction with the most impact in the yield is highlighted in red. (A)** Results for the glycerol pathway. According to the coefficients, the reaction with the most impact is the glycerol-3-phosphate dehydrogenase (G3pD), which due to its positive FCC is a potential target for overexpression; **(B)** Results for the malonyl-CoA pathway. The results showed that the acetyl-CoA carboxylase (AccC) is a potential bottleneck in the pathway due to the positive FCC; hence, another target for overexpression; **(C)** Results for the *β*-alanine pathway. The highest FCC was for the aspartate aminotransferase (AspAT) which appears to be an ideal target for an overexpression.

which corresponds to a 1.5-fold higher yield than Mutant Glu-Mcoa 0 (Fig 9B). Unfortunately, once again, a second iteration revealed that the model was unable to reach a steady-state. Thus, it was not possible to identify other potential targets using this methodology.

Regarding the *β*-alanine pathway, the MCA showed that the model has several reactions affecting the AA yield. However, the reaction with the most significant impact is catalysed by AspAT enzyme (Fig 8C). This reaction allows converting oxaloacetate and *L*-glutamate into aspartate, which is in turn converted to *β*-alanine. Moreover, reports from the literature suggest that increasing the bioavailability of aspartate leads to a higher 3-HP production, as *in*

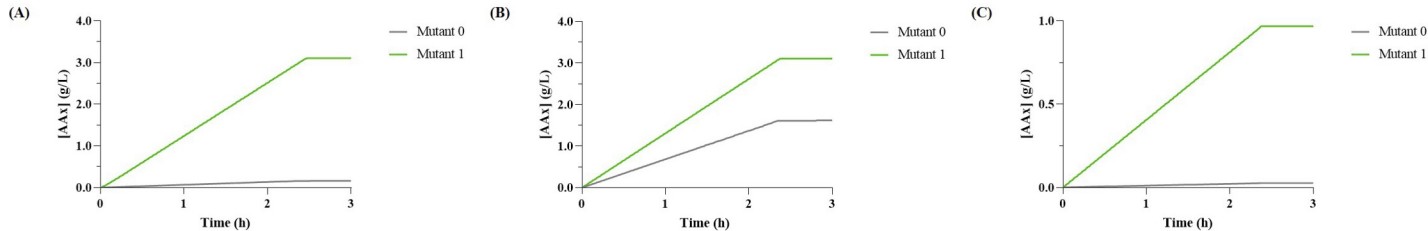

**Fig 9. Comparison between the original acrylic acid (AA) production with the results obtained for the mutants developed with the optimisation strategies identified. (A)** AA production using glycerol pathway. Mutant 0 represents the model with the heterologous pathway, and Mutant 1 the same model with a 45-fold increase in the $V_{max}$ of the glycerol-3-phosphate dehydrogenase (G3pD) reaction; **(B)** AA production using malonyl-CoA pathway. Mutant 0 represents the model with the heterologous pathway, and Mutant 1 the same model with a 2.5-fold increase in the $V_{max}$ of the acetyl-CoA carboxylase (AccC); **(C)** AA production for the $\beta$-alanine pathway. Mutant 0 represents the model with the heterologous pathway, and Mutant 1 the same model with a 50-fold increase in the $V_{max}$ of the aspartate aminotransferase (AspAT).

*vivo* experiments with *Saccharomyces cerevisiae* and *E. coli* showed it as a viable optimisation strategy [11,20]. Moreover, the resulting coefficient was positive, which indicates that it is a potential bottleneck impairing the downstream flux towards the heterologous pathway; thus, the optimisation goal was to overexpress this enzyme. COPASI estimated a 50-fold overexpression for maximising the production yields, resulting in a $V_{max}$ of 127.4869 mM/s. It is worth noting that, even though this value is significantly higher than what is biologically feasible, the goal of the optimisation was to identify potential targets and not meticulously predict the final $V_{max}$ value. With this change, the predicted AA production was 0.97 g/L, which corresponds to a 28-fold increase (Fig 9C). A subsequent MCA revealed that the main limiting factor to AA production in Mutant Glu-Ba 1 was the amount of glucose provided to the model; hence the optimisation was terminated with only one target identified. Similarly, the AspC gene, which is responsible for the production of $\beta$-alanine, can also be considered as a limiting factor for pathway flux, thus becoming a target for optimisation, as the $V_{max}$ was increased for the $\beta$-alanine model to work correctly (S1 Appendix, section 1.2.1).

The goal of these optimisations was to provide guidelines that can be later implemented *in vivo* and not predict the AA production accurately. Nonetheless, the final concentrations obtained with the new mutants were compared with previous results. As shown in Table 4, the same concentration of AA (3.11 g/L) was produced by the glycerol and malonyl-CoA pathways. The $\beta$-alanine pathway also showed a substantial yield increase (0.97 g/L). However, the value is still considerably lower than the obtained with remaining pathways. It is important to notice that the four targets suggested by this analysis aim at increasing the bioavailability of the intermediaries (glycerol, malonyl-CoA, or $\beta$-alanine). Nevertheless, these models only comprise the CCM. Thus, other strategies to force additional flux towards the heterologous pathway may also prove useful.

**Table 4. Summarised results of acrylic acid production for the mutant strains developed for the glycerol, malonyl-CoA, and $\beta$-alanine models, using 10 g/L of glucose as substrate.**

| Pathway | Strain | Predicted Titer (g/L) |
|---|---|---|
| Glycerol | Mutant 0 | 0.16 |
| | Mutant 1 | 3.11 |
| Malonyl-CoA | Mutant 0 | 1.62 |
| | Mutant 1 | 3.11 |
| $\beta$-alanine | Mutant 0 | 0.026 |
| | Mutant 1 | 0.97 |

## Conclusion

In conclusion, these models seem to be more accurate in predicting 3-HP synthesis as the $V_{max}$ for the heterologous enzymes was calculated in excess, not to limit the reaction flux. The models exhibited limitations regarding the assimilation of glycerol and $\beta$-alanine production. An effort was put forward towards finding proteomics data that included the absolute quantification of such enzymes to solve these problems. However, to the best of our knowledge, no quantification data was found; therefore, it will be important in the future to seek such data or, in the lack of new data, to determine it experimentally. Moreover, this analysis indicates that, even though not exhibiting the highest yields, the malonyl-CoA path appears to be the best choice for industrial-scale production of AA, as it does not require vitamin supplementation and there is still room for optimisation. As for the comparison between glucose and glycerol, an overall best carbon source does not emerge from this work. Instead, the answer is specific to the selected pathway. Finally, this study also suggests four optimisation targets that, theoretically, should result in higher yields. Nonetheless, as this study only focused on the computational work, validation with *in vivo* experiments is required in the future to further confirm these results.

## Materials and methods

### Kinetic modelling

The dynamic model developed by Millard et al. (2017) [21] was used as a chassis to insert the heterologous pathways. However, the model did not include the production of glycerol, malonyl-CoA and $\beta$-alanine, which are naturally produced in *E. coli*. Therefore, the first step was to extend the original model to include these metabolites. Subsequently, each heterologous pathway was added separately to compare 3-HP and AA synthesis.

**Parameter selection.** Kinetic equations and their respective parameters were retrieved from the available literature. Databases like BioCyc [22], BRENDA [23], Sabio-RK [24] and eQuilibrator [25] were used to identify the kinetic mechanism of each enzyme and obtain their respective parameters. Furthermore, instead of a single value, the average of all parameters found for each enzyme, excluding outliers, was used to obtain a better representation. A summary of the values considered when calculating the mean value for each reaction is presented in S2 Appendix.

Regarding the parameters required to describe a reaction, the maximal rate ($V_{max}$) is usually not reported in the literature. Unfortunately, this parameter is highly dependent on the specificity of the assay conditions. Hence, the specific activity or the turnover (a.k.a. $K_{cat}$) are reported instead. Two distinct methods were used, as a workaround, to estimate values for this parameter. Method 1, adapted from the work of Chassagnole and colleagues [26], was used for reactions belonging to *E. coli*'s native metabolism. Initially, a steady-state flux distribution is determined for the original kinetic model, using the default settings of the COPASI software [27] steady-state task and not further fitted to the expected behaviour. Then, a genome-scale model of *E. coli* K-12 MG1655 (in this case iML1515) [28] was used to predict the flux of the new reactions. For this purpose, the common reactions between the kinetic model and the stoichiometric model are constrained to the previously determined flux distribution ($\pm0.01$ mM/s). Then, a flux variability analysis [29] was performed to estimate the maximum flux ($v$) of the desired reaction for the given constraints. Then, by equalising $v$ to the respective enzyme rate law ($V_{max} \times F(X,K)$), the following equation is obtained:

$$v = V_{max} \cdot F(X, K) \Leftrightarrow V_{max} = \frac{v}{F(X, K)} \tag{1}$$

in which $X$ is a vector of parameters, and $K$ a vector of steady-state concentrations for the metabolites involved in the respective reaction. Furthermore, notice that for newly added metabolites, the steady-state concentration was assumed to be 1 mM. The resulting $V_{max}$ values are presented in the S2 Table.

Method 2 was used for reactions of the heterologous pathways. In this method, the $V_{max}$ was estimated assuming that the total concentration of enzyme was in surplus (100 mM), thus calculating this parameter as shown in Eq 2:

$$V_{max} = K_{cat} \cdot [E]_T \qquad (2)$$

Even though an enzyme concentration of 100 mM is beyond what is biologically feasible, this value was selected to avoid creating artificial bottlenecks that would impair this analysis. Furthermore, the authors also tested different concentrations to assess the impact on AA production and the results are presented in S1 Appendix, section 1. and Figs 3 and S6 and S7.

**Extension of the Central Carbon Metabolism (CCM).** The production of glycerol, malonyl-CoA, and $\beta$-alanine had to be included in the model to insert the three heterologous pathways (Fig 10). The reactions catalysed by the glycerol-3-phosphate dehydrogenase (G3pD) and glycerol-3-phosphate phosphatase (G3pP) enzymes are required to produce glycerol from dihydroxyacetone phosphate. Considering that this route has to be reversible to use glycerol as a carbon source, reactions catalysed by the glycerol kinase (GlyK), glycerol dehydrogenase (GlyD) and the dihydroxyacetone phosphate transferase (DhaPT) enzymes were included too (Fig 10). As shown in Fig 10, one reaction is required to obtain malonyl-CoA, namely the reaction catalysed by the acetyl-CoA carboxylase (AccC) enzyme. Finally, three reactions were included for the $\beta$-alanine pathway. Two of these, promoted by the aspartate aminotransferase (AspAT) and the aspartate carboxylase (AspC) enzymes, are required for the production of $\beta$-alanine. A reaction, catalysed by the *L*-glutamate dehydrogenase (GluD) enzyme, is used to produce glutamate, which is required by the AspAT to produce aspartate (Fig 10) [22,30].

All reactions and respective stoichiometry are shown below:

$$\textbf{G3pD}: \; \text{Dihydroxyacetone phosphate} + \text{NADPH} + \text{H}^+ = \text{Glycerol}-3-\text{phosphate} + \text{NADP}^+ \quad (3)$$

$$\textbf{G3pP}: \; \text{Glycerol}-3-\text{phosphate} + \text{H}_2\text{O} \rightarrow \text{Glycerol} + \text{P}_i \qquad (4)$$

$$\textbf{GlyK}: \; \text{Glycerol} + \text{ATP} \rightarrow \text{Glycerol}-3-\text{phosphate} + \text{ADP} + \text{H}^+ \qquad (5)$$

$$\textbf{GlyD}: \; \text{Glycerol} + \text{NAD}^+ \rightarrow \text{Dihydroxyacetone} + \text{NADH} + \text{H}^+ \qquad (6)$$

$$\textbf{DhaPT}: \; \text{Dihydroxyacetone} + \text{Phosphoenolpyruvate} \rightarrow \text{Dihydroxyacetone phosphate} + \text{Pyruvate} \quad (7)$$

$$\textbf{AccC}: \; \text{Acetyl}-\text{CoA} + \text{ATP} + \text{HCO}_3 \rightarrow \text{Malonyl}-\text{CoA} + \text{ADP} + \text{P}_i \qquad (8)$$

$$\textbf{GluD}: \; \alpha-\text{Ketoglutarate} + \text{NADPH} + \text{NH}_4{}^+ \rightarrow L-\text{Glutamate} + \text{NADP}^+ + \text{H}_2\text{O} \qquad (9)$$

$$\textbf{AspAT}: \; \text{Oxaloacetate} + L-\text{Glutamate} = L-\text{Aspartate} + \alpha-\text{Ketoglutarate} \qquad (10)$$

$$\textbf{AspC}: \; \text{Aspartate} \rightarrow \beta-\text{alanine} + \text{CO}_2 \qquad (11)$$

Furthermore, the kinetic law equation and the respective parameters for each of the previously described reactions are presented in Table 5.

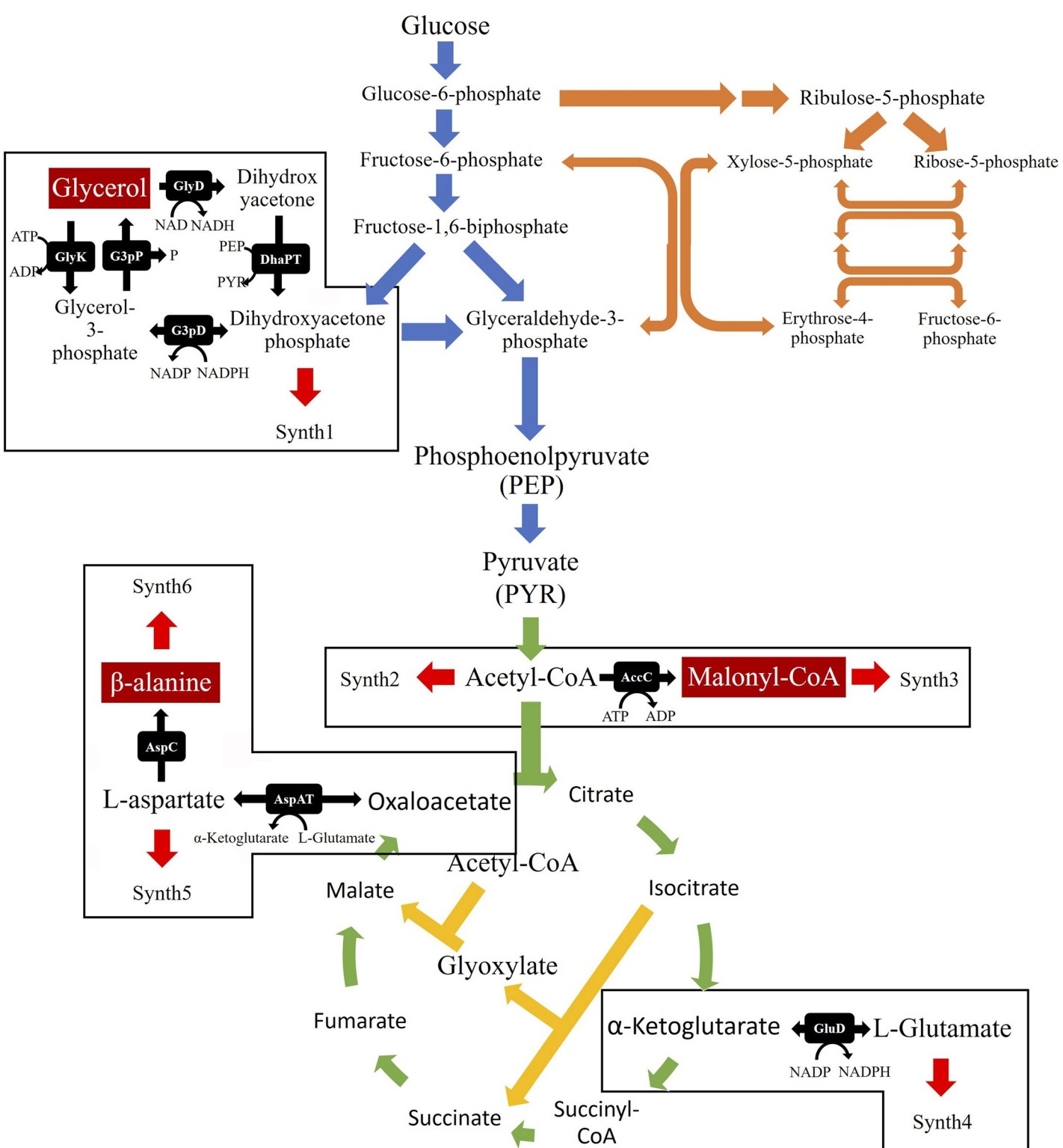

**Fig 10. Representation of the central carbon metabolism of *Escherichia coli* and the reactions added to the kinetic model.** The reactions depicted by the blue, orange, green and yellow arrows represent, respectively, the glycolysis, pentose-phosphate pathway, tricarboxylic acid cycle and the glyoxylate shunt, which are all present in the original model. The black arrows represent the nine reactions that were added to the model. Finally, red arrows depict the Synth reactions added to account for the presence of the newly added metabolites in other pathways.

**Table 5. Rate Law (RL) equations, kinetic parameters and the respective references for each reaction that belong to the native metabolism of *Escherichia coli*.**

| Reaction | E.C. number | RL | Equation | Parameters | Reference |
|---|---|---|---|---|---|
| G3pD | 1.1.1.94 | Rapid Equilibrium Random Bi Bi; | $V_{max} \cdot \dfrac{\left(\dfrac{A \cdot B - \left(\frac{P \cdot Q}{K_{eq}}\right)}{K_{m,a} \cdot K_{m,b}}\right)}{\left(1 + \frac{A}{K_{m,a}} \cdot \frac{B}{K_{m,b}}\right) + \left(1 + \frac{P}{K_{m,p}} \cdot \frac{Q}{K_{m,q}}\right) - 1}$ | $K_{m,a} = 0.175$ mM; $K_{m,b} = 0.0037$ mM $K_{m,p} = 0.12$ mM; $K_{m,q} = 0.165$ mM $K_{eq} = 900$ | [32–34] |
| G3pP | 3.1.3.21 | Michaelis-Menten | $V_{max} \cdot \dfrac{A}{K_m + A}$ | $K_m = 2.9$ mM | [35] |
| GlyK | 2.7.1.30 | Random Bi Bi | $\dfrac{V_{max} \cdot A \cdot B}{K_{d,a} \cdot K_{m,b} + K_{m,b} \cdot A + K_{m,a} \cdot B + A \cdot B}$ | $K_{m,a} = 0.0084$ mM; $K_{m,b} = 0.0049$ mM $K_{d,a} = 0.086$ mM | [36–39] |
| AccC | 2.1.3.15 | Order Bi Bi | $V_{max} \cdot \dfrac{A}{K_{m,A} \cdot \left(1 + \frac{P}{K_{i,P}}\right) + A} \cdot \dfrac{B}{K_{m,B} + B}$ | $K_{m,a} = 0.018$ mM; $K_{m,b} = 0.06$ mM $K_{i,p} = 0.07$ mM | [40–42] |
| GluD | 1.4.1.4 | Michaelis-Menten | $V_{max} \cdot \dfrac{A}{K_{m,a} + A} \cdot \dfrac{B}{K_{m,b} + B}$ | $K_{m,a} = 0.495$ mM; $K_{m,b} = 0.037$ mM | [43–46] |
| AspAT | 2.6.1.1 | Ping-Pong Bi Bi | $V_{max} \cdot \dfrac{\left(\dfrac{A \cdot B - \left(\frac{P \cdot Q}{K_{eq}}\right)}{K_{m,a} \cdot K_{m,b}}\right)}{\left(1 + \frac{A}{K_{m,a}} \cdot \frac{Q}{K_{m,q}}\right) \cdot \left(1 + \frac{B}{K_{m,b}} \cdot \frac{P}{K_{m,p}}\right)}$ | $K_{m,a} = 19.07$ mM; $K_{m,b} = 0.19$ mM $K_{m,p} = 0.437$ mM; $K_{m,q} = 2.94$ mM $K_{eq} = 3.2$ | [47–51] |
| AspC | 4.1.1.11 | Michaelis-Menten | $V_{max} \cdot \dfrac{A}{K_m + A}$ | $K_m = 0.155$ mM | [52,53] |
| GlyD | 1.1.1.6 | Hill Cooperativity | $V_{max} \cdot \dfrac{A^n}{(K_{m,a})^n + B^n} \cdot \dfrac{B^n}{(K_{m,b})^n + B^n}$ | $K_{m,a} = 47.83$ mM; $K_{m,b} = 1.385$ mM $n = 0.98$ | [54,55] |
| DhaPT | 2.7.1.121 | Mass Action | $k \cdot A \cdot B$ | Not Found | - |

An additional set of pseudo-reactions was included in the model; the Synth reactions (Fig 10). These reactions, inspired by the work of Chassagnole et al. (2002) [26] and Machado et al. (2014) [31], are used to represent the pathways involved in the breakdown of the newly added metabolites. Mass action kinetics was assumed for these reactions and, using the same principle as Method 1, the sum of all fluxes from the reactions that metabolise each metabolite in the stoichiometric model was used to determine the *k* values for each synth reaction. The resulting *k* values are available in the S3 Table.

**Pathways for acrylic acid production.** The following step was to insert the three heterologous pathways to produce AA separately into the extended CCM model. All pathways encompass two different phases, the production of an intermediary compound, namely 3-HP, and subsequent production of AA (Fig 1).

The first phase involves two different enzymes in each pathway. Regarding the glycerol pathway, such enzymes are the glycerol dehydratase (GlyDH) and the 3-hydroxypropionaldehyde dehydrogenase (3hpaD). In the malonyl-CoA pathway, the malonyl-CoA reductase (McoaR) enzyme is responsible for the production of malonic semialdehyde (MSA), which is then converted into 3-HP by the malonic semialdehyde reductase (MsaR). Regarding the β-alanine pathway, the *β*-alanine aminotransferase (BaAT) enzyme promotes the conversion of β-alanine together with *α*-ketoglutarate into *L*-glutamate and MSA. The latter is then converted into 3-HP by the MsaR.

$$\textbf{GlyDH}: \text{Glycerol} \rightarrow 3-\text{HPA} + H_2O \tag{12}$$

$$\textbf{3hpaD}: 3-\text{HPA} + NAD^+ + H_2O \rightarrow 3-\text{HP} + NADH + 2H^+ \tag{13}$$

$$\textbf{McoaR}: \text{Malonyl}-\text{CoA} + NADPH + H^+ \rightarrow MSA + CoA + NADP^+ \tag{14}$$

$$\textbf{MsaR}: MSA + NADPH + H^+ \rightarrow 3-\text{HP} + NADP^+ \tag{15}$$

$$\textbf{BaAT}: \beta-\text{alanine} + \alpha-\text{Ketoglutarate} \rightarrow \text{MSA} + L-\text{Glutamate} \qquad (16)$$

The final step is to convert the newly formed 3-HP into AA. This process involves the production of 3-hydroxypropionyl-CoA (3-HP-CoA) by the 3-hydroxypropionyl-CoA synthase (3hpcoaS), the subsequent formation of acrylyl-CoA (AA-CoA) by the 3-hydroxypropionyl-CoA dehydratase (3hpcoaDH), and finally, the production of AA by the acrylyl-CoA thioesterase (AcoaTioE) enzyme, as shown in Fig 1. The stoichiometry of these reactions is likewise shown below, and the respective kinetic parameters presented in Table 6.

$$\textbf{3hpcoaS}: 3-\text{HP} + \text{CoA} + \text{ATP} \rightarrow 3-\text{HP}-\text{CoA} + 2\text{P}_i + \text{AMP} \qquad (17)$$

$$\textbf{3hpcoaDH}: 3-\text{HP}-\text{CoA} \rightarrow \text{AA}-\text{CoA} + \text{H}_2\text{O} \qquad (18)$$

$$\textbf{AcoaTioE}: \text{AA}-\text{CoA} + \text{H}_2\text{O} \rightarrow \text{AA} + \text{CoA} + \text{H}^+ \qquad (19)$$

Four models were created for each of the three pathways resulting in a total of twelve distinct models. To be more precise, for each pathway, models to produce 3-HP or AA, from glucose or glycerol as carbon sources, were put forward. All models can be found at the Biomodels database, and the respective ID is available in S1 Table.

## Time course simulation

Time course simulations were performed to assess 3-HP and AA production over time, using the deterministic method (LSODA) from COPASI [27], with a duration of three or six hours, to allow the consumption of all available carbon source. Since the available carbon sources have a different number of carbons, the initial concentration of such molecules had to ensure that the amount of carbon provided to the model was the same. Thus, the initial concentrations for glucose and glycerol were 55.5 mM (10 g/L) and 217.2 mM (20 g/L), respectively,

**Table 6. Rate Law (RL) equations, kinetic parameters and the respective references for each reaction of the three heterologous pathways (glycerol, malonyl-CoA and $\beta$-alanine) required to produce acrylic acid.**

| Reaction | E.C. number | RL | Equation | Parameters | Reference |
|---|---|---|---|---|---|
| GlyDH | 4.2.1.28 | Specific Activation | $\frac{E \cdot K_{cat} \cdot A \cdot Activator}{K_{m,a} \cdot K_a + (K_{m,a} + A) \cdot Activator}$ | $K_{cat} = 0.0621$ s$^{-1}$; $K_m = 6.15$ mM $K_a = 0.008$ mM | [56,57] |
| 3hpaD | 1.2.1.99 | Michaelis-Menten | $E \cdot K_{cat} \cdot \frac{A}{K_{m,a} \cdot \left(1 + \frac{P}{K_{i,p}}\right) + A} \cdot \frac{B}{K_{m,b} + B}$ | $K_{cat} = 16.73$ s$^{-1}$; $K_{m,a} = 0.39$ mM $K_{m,b} = 1.3$ mM; $K_{i,p} = 0.12$ | [58] |
| McoaR | 1.2.1.75 | Michaelis-Menten | $\frac{E \cdot K_{cat} \cdot A \cdot B}{K_{m,a} \cdot K_{m,b} + K_{m,b} \cdot A + K_{m,a} \cdot B + A \cdot B}$ | $K_{cat} = 50$ s$^{-1}$; $Km_a = 0.3$ mM $K_{m,b} = 0.03$ mM | [59,60] |
| MsaR | 2.6.1.19 | Michaelis-Menten | $\frac{E \cdot K_{cat} \cdot A \cdot B}{K_{m,a} \cdot K_{m,b} + K_{m,b} \cdot A + K_{m,a} \cdot B + A \cdot B}$ | $K_{cat} = 115$ s$^{-1}$; $K_{m,a} = 0.07$ mM $K_{m,b} = 0.07$ mM | [61] |
| BaTA | 1.1.1.298 | Ping-Pong Bi Bi | $\frac{E \cdot K_{cat} \cdot A \cdot B}{K_{m,b} \cdot A + K_{m,a} \cdot B \cdot \left(1 + \frac{B}{K_{i,B}}\right) + A \cdot B}$ | $K_{cat} = 47.4$ s$^{-1}$; $K_{m,a} = 5.8$ mM $K_{m,b} = 1.07$ mM; $Ki_b = 10.2$ mM | [62] |
| 3hpcoaS | 6.2.1.36 | Michaelis-Menten | $E \cdot K_{cat} \cdot \frac{A}{K_{m,a} + A} \cdot \frac{B}{K_{m,b} + B} \cdot \frac{C}{K_{m,c} + C}$ | $K_{cat} = 36$ s$^{-1}$; $K_{m,a} = 0.015$ mM $K_{m,b} = 0.01$ mM; $K_{m,c} = 0.05$ mM | [63] |
| 3hpcoaDH | 4.2.1.116 | Michaelis-Menten | $\frac{E \cdot K_{cat} \cdot A}{K_m + A}$ | $K_{ca\ t} = 96$ s$^{-1}$; $K_m = 0.06$ mM | [64] |
| AcoaTioE | 3.1.2.20 | Michaelis-Menten | $\frac{E \cdot K_{cat} \cdot A}{K_m + A}$ | $K_{ca\ t} = 0.55$ s$^{-1}$; $K_m = 0.167$ mM | [65] |

The enzyme concentration ($E$) value used for all the reactions was 100 mM.

which allowed comparing the three pathways for each carbon source. The model was assessed to available literature regarding these pathways, in which the simulations' initial concentration of the carbon source was set to replicate the initial conditions of published results.

## Optimisation strategies

The first step was to determine the flux control coefficients (FCC), through a metabolic control analysis (MCA). Initially, a feed and drains for glycerol, malonyl-CoA, $\beta$-alanine, and AA were included to replicate a continuous model and find a valid steady-state, which is required to determine the FCCs. These coefficients reflect the level of control that each reaction has over the formation of AA. The optimisation was then performed, using the automated optimisation tool provided by COPASI [27], for the initial models with only an initial concentration of glucose (10 g/L) and without drains for the end metabolites. Here, the algorithm proposed new *in silico* mutant strains, in which the reaction with most influence was either over-expressed, under-expressed, or knocked-out through a change in the $V_{max}$, according to the respective coefficient. The goal of the optimisation was not to meticulously predict the final concentration of AA, but rather to find promising targets for optimisation. Hence, the changes in the $V_{max}$ were limited to 50 times the original value to allow overcoming the influence of such reaction *in silico*, while not impairing the *in vivo* implementation.

The objective function was the maximisation of AA concentration in the time course task. After creating new mutants, the process was repeated to optimise the mutant strains further. The task eventually stopped when either the glucose feed was limiting the production of AA, the limiting reaction was already optimised, or the system could no longer reach a stable steady-state point during the MCA.

## Supporting information

**S1 Fig. Comparison of the flux distribution of the central carbon metabolism (CCM) from glycerol between the extended dynamic model and experimentaly measured values.** (A) Steady-State flux distribution from glycerol obtained from the extended kinetic model of *E.coli*'s CCM; (B) Flux distribution from glycerol obtained experimentaly by Toya et al. (2018) [12]; (C) Flux distribution from glycerol obtained experimentaly by Yao et al. (2019) [13].
(TIF)

**S2 Fig. Variation of β-alanine (BA) production over time.** (A) β-alanine concentration using the $V_{max}$ for the aspartate carboxylase (AspC) enzyme calculated using Method 1 ($1.15 \times 10^{-05}$ mM/s). (B) β-alanine concentration using the Vmax for the AspC calculated using Method 2 (57 mM/s).
(TIF)

**S3 Fig. Results of the time course simulations in the original glycerol model.** (A) Glycerol (GLY) consumption. (B) Production of 3-hydroxypropionate (3-HP). (C) Acrylic acid (AA) production. (D) Flux of the glycerol dehydrogenase (GlyD).
(TIF)

**S4 Fig. Results of the time course simulations in the original glycerol model after the affinity towards NAD+ of the GlyD was changed to 0.0165 mM.** (A) Glycerol (GLY) consumption. (B) Production of 3-hydroxypropionate (3-HP). (C) Acrylic acid (AA) production. (D) Flux of the glycerol dehydrogenase (GlyD).
(TIF)

**S5 Fig. Results of the time course simulations in the original glycerol model after the $V_{max}$ of the reaction GlyD was changed to 4298.4 mM/s.** (A) Glycerol (GLY) consumption. (B) Production of 3-hydroxypropionate (3-HP). (C) Variation of dihydroxyacetone (DHA) concentration. (D) Acrylic acid (AA) production.
(TIF)

**S6 Fig. Impact of enzyme concentration in the acrylic acid (AA) producing models.** Time course simulation of AA production from glucose and fluxes of the heterologous reactions when using different enzyme concentrations to determine the $V_{max}$ value, according to method 2, for the glycerol route (A), malonyl-CoA route (B), and $\beta$-alanine route (C). Three concentrations were simulated: 100 mM (orange lines), 10 mM (green lines), and 1 mM (blue line)
(TIF)

**S7 Fig. Impact of enzyme concentration in the acrylic acid (AA) producing models.** Time course simulation of AA production from glycerol and fluxes of the heterologous reactions when using different enzyme concentrations to determine the $V_{max}$ value, according to method 2, for the glycerol route (A), malonyl-CoA route (B), and $\beta$-alanine route (C). Three concentrations were simulated: 100 mM (orange lines), 10 mM (green lines), and 1 mM (blue line)
(TIF)

**S1 Table. Details of twelve kinetic models developed to achieve 3-hydroxypropionate (3-HP) and acrylic acid (AA) from either glucose or glycerol.**
(XLSX)

**S2 Table. $V_{max}$ values calculated for the reactions required for the extension of the central carbon metabolism.**
(XLSX)

**S3 Table. Synth reactions added to the model and respective parameters.** These reactions were created for dihydroxyacetone phosphate (DAP), acetyl-CoA (ACCOA), malonyl-CoA (MCOA), L-glutamate (LGLU), L-aspartate (ASP) and $\beta$-alanine (BA).
(XLSX)

**S1 Appendix. Supplementary results and parameter adjustments.** Additional results, explanations behind parameter adjustments adopted to circumvent simulation issues, and results for the $V_{max}$ calculation using method 1 and method 2.
(PDF)

**S2 Appendix. Kinetic parameters.** This file presents all the kinetic parameters and equations used to model AA production.
(PDF)

## Author Contributions

**Conceptualization:** Joana Rodrigues, Lígia Rodrigues, Oscar Dias.

**Data curation:** Alexandre Oliveira, Joana Rodrigues, Oscar Dias.

**Formal analysis:** Alexandre Oliveira, Oscar Dias.

**Investigation:** Alexandre Oliveira, Joana Rodrigues, Oscar Dias.

**Methodology:** Alexandre Oliveira, Joana Rodrigues, Eugénio Campos Ferreira, Lígia Rodrigues, Oscar Dias.

**Resources:** Eugénio Campos Ferreira.

**Software:** Alexandre Oliveira.

**Supervision:** Joana Rodrigues, Lígia Rodrigues, Oscar Dias.

**Validation:** Alexandre Oliveira, Joana Rodrigues, Oscar Dias.

**Visualization:** Alexandre Oliveira, Oscar Dias.

**Writing – original draft:** Alexandre Oliveira.

**Writing – review & editing:** Alexandre Oliveira, Joana Rodrigues, Eugénio Campos Ferreira, Lígia Rodrigues, Oscar Dias.

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
