## [Decision Letter · Decision Letter 0]

20 Aug 2020

Dear Dr. Dias,

Thank you very much for submitting your manuscript "A kinetic model of the central carbon metabolism for acrylic acid production in Escherichia coli" for consideration at PLOS Computational Biology.

As with all papers reviewed by the journal, your manuscript was reviewed by members of the editorial board and by several independent reviewers. In light of the reviews (below this email), we would like to invite the resubmission of a significantly-revised version that takes into account the reviewers' comments.

We cannot make any decision about publication until we have seen the revised manuscript and your response to the reviewers' comments. Your revised manuscript is also likely to be sent to reviewers for further evaluation.

Sincerely,

Pedro Mendes, PhD

Associate Editor

PLOS Computational Biology

Jason Papin

Editor-in-Chief

PLOS Computational Biology

Reviewer's Responses to Questions

**Comments to the Authors:**

Reviewer #1: the review is uploaded as an attachment

Reviewer #2: Reproducibility report has been uploaded as an attachment.

Reviewer #3: In this work, the authors extend an existing kinetic model of E. coli metabolism to study the production of acrylic acid. They assess three production pathways for the product: glycerol, malonyl-CoA, and beta-alanine, using either glucose or glycerol as the primary carbon source depending on which pathway was used. The authors then predict targets for overexpression to improve production.

The effort is interesting and modeling approaches using kinetic models are worthwhile for improving our understanding of strain designs. I have concerns about the execution and validation of predictions that may limit the impact of the work however.

In general, the authors could do more to build confidence in the model predictions. The details of the model construction, parameterization, and validation are not presented in the forefront of the Results section. There are a number of titer predictions which are used as validation initially, which are in excellent agreement for certain conditions although not others (Table 2). However, it is not clear how the model was set up to obtain these results. The methods to debug the model presented in the Supplementary Text seemed ad hoc and it was not clear if the final models were free of issues, as for example an independent validation data set would help to verify. In the end, several models were not able to be evaluated due to issues with stability or non-physiological results. Additionally, the methods for gapfilling v_max values seem confusing or problematic, as my comments below discuss in more detail. Additional upfront discussion of these issues, how they were overcome, and why the model can still be trusted would help the reader to have confidence in the model.

The optimization effort in the work seems promising but plagued by further issues of model instability, which casts doubt on the robustness of the methodology. It does not appear that any validation of overexpression targets from comparison to prior work was done, if any such data is available. Thus, it is not clear how accurate or valuable these predictions are. Nonetheless, if these targets could be experimentally validated, it would be of considerable interest.

I would be supportive of publishing this effort if the authors can provide additional clarity on their methods, investigation into parameter sensitivity (especially related to the numerous unstable models) more globally, and additional validation of the model results. In its current form, the manuscript provides more questions than answers in many places.

Major comments

- The methods for determining initial conditions of the models are especially opaque – I can hardly find any details at all of how the initial steady state of the model was determined. This has direct implications to whether the agreement between data and model titers in Table 2 was fit by the authors or emergent from model behavior. The latter is obviously more impressive, but fitting to expected behavior is fine as well, as long as some type of model validation is presented elsewhere.

- Method 1 for parameterization, which uses an estimate of the kinetic v_max calculated by Flux Variability Analysis, is problematic inherently. The v_max term in FVA and the Michaelis-Menten kinetic v_max term are completely different fundamentally. The former v_max is a theoretical value having to do with the constraints on the system’s mass balances given measured metabolic exchanges and a growth rate. For many reactions, v_max solved with FVA may even hit ‘infinity’ due to loops in the network, or may be impractically high because of a lack of known constraints on the pathway. Although they share a name, the FVA v_max is very different then the kinetic v_max, which is simply the maximum rate that a certain mass of enzyme can catalyze a reaction, approximated by E_tot*k_cat. To justify the use of Method 1 for estimating the latter v_max parameters, the authors need to perform validation by reproducing measured kinetic v_max values with their method, i.e. extend their S1 Table 2 calculations to reactions with measured v_max and compare. My suspicion is that the authors would be better off inserting a physiological flux value rather than an FVA-calculated maximum flux value, for example as has been done in previous work for estimating enzyme turnover rates (see PMID: 26951675).

- I also do not understand the authors’ justification for Method 2. They state that in cases where the heterologous enzyme is high, then v_max = k_cat * E_tot. In a simple Michaelis-Menten derivation, then this equation is the definition of the v_max term, so I don’t know why the enzyme concentration is relevant. If the authors are saying the v = v_max at high enzyme levels, then this seems problematic from the factor that at very high enzyme concentrations (100mM as they state), the substrate is not likely to be high enough concentration to saturate enzyme sites, so v would be much less than v_max. Hopefully the authors can clarify better how Method 2 works and better justify it as necessary.

- In general, it would be nice if the model predictions were presented more systematically in some format. The existing tables are quite nice, but the models’ many predictions and failure modes are discussed haphazardly throughout the manuscript which makes it a bit difficult to summarize the designs of greatest excitement. Tables 2 and 4 do a good job, but even more summary like this would be nice if possible.

Reviewer #4: after reading the manuscript, I find it lacks sufficient novelty a good study requires, only an extension of the work by Millard et al. 2017. there is key difference between the two studies. Millard et al. 2017 used the Core carbon metalism, which is well defined and visualized as opposed to this study, to show the importance of metabolic regulation rather than some specific cases, while this study uses their method for studying a specific functional objective, the production of the AA. it is almost impossible and difficulties remain. the authors recognize some of them and made some workarounds which to me do not work adequately.

First, the model should be clearly presented as how many reactions and metabolites and what the constraints are. all these are lacking in the study.

Second, these three pathways work in a network which includes three of them operating simultaneously (I suppose or the authors should indicate otherwise), so they cannot be modeled as individual pathways as they were in the study.

Third, for this kind study, some kind of experimental validation is required.

Fourth, Kcat for Vmax and the parameters estimated from FBA of a global model for a subnetwork are questionable, although I understand these parameters may be not available otherwise.

l64-69: the aerobic/anaerobic conditions should be presented for 3PH production. three routes should be compared in terms of energy consumption. this also makes a whole world of difference in simulations.

l72-74: why glucose and glycerol are capitalized in the first letter? why the second sentence is relevant here?

l90: the simulations should be introduced very briefly as this section follows an introduction, skipping the Methods section.

**Have all data underlying the figures and results presented in the manuscript been provided?**

Reviewer #1: **No: **All models are available at https://cutt.ly/aaKineticModels. However, this address appears to refer to an URL shortener service, which cannot guarantee their long-term availability. Models should be deposited to dedicated repositories (such as biomodels database) to ensure long-term availability.

Reviewer #2: Yes

Reviewer #3: Yes

Reviewer #4: Yes

PLOS authors have the option to publish the peer review history of their article (what does this mean?). If published, this will include your full peer review and any attached files.

Reviewer #1: No

Reviewer #2: **Yes: **Anand K. Rampadarath

Reviewer #3: No

Reviewer #4: No
---

## [Decision Letter · Decision Letter 1]

12 Jan 2021

Dear Dr. Dias,

We are pleased to inform you that your manuscript 'A kinetic model of the central carbon metabolism for acrylic acid production in Escherichia coli' has been provisionally accepted for publication in PLOS Computational Biology.

Best regards,

Pedro Mendes, PhD

Associate Editor

PLOS Computational Biology

Jason Papin

Editor-in-Chief

PLOS Computational Biology

Reviewer's Responses to Questions

**Comments to the Authors:**

Reviewer #1: I have carefully read the revised manuscript, and most of the comments have been addressed adequately, which has significantly improved the manuscript. I would like to thank the authors for depositing the models in biomodels database.

Reviewer #3: I appreciate the additions made by the authors to clarify the results and methods of the manuscript. However, I am unclear on why the authors still have not performed the validation of v_max estimates from FVA that appears to me to be necessary given my previous arguments. Citing a 20 year old paper that used a similar method (also without validation) is not sufficient to demonstrate that this method produces realistic v_max estimates. Some sort of cross-validation approach to validate their predictions against known v_max values (or kcat values, given known enzyme concentrations) within central metabolism would be much more valuable than a citation, and seems simple to test. As the work still does not contain any experimental validation and derives from an existing kinetic model, it is critical that it makes clear theoretical advances, and validating a parameterization workflow would be one such advance. I would like to be more supportive of the work but remaining modeling issues should be resolved conclusively if the authors do not want to experimentally validate their predictions. In that vein, I would also be supportive of the authors resolving the reversibility issues highlighted by other reviewers through an approach like parameter sampling as well.

Reviewer #4: all my comments are addressed.

**Have all data underlying the figures and results presented in the manuscript been provided?**

Reviewer #1: Yes

Reviewer #3: Yes

Reviewer #4: None

PLOS authors have the option to publish the peer review history of their article (what does this mean?). If published, this will include your full peer review and any attached files.

Reviewer #1: No

Reviewer #3: No

Reviewer #4: No

---

## [Editor Report · Acceptance letter]

2 Mar 2021

PCOMPBIOL-D-20-01252R1 

A kinetic model of the central carbon metabolism for acrylic acid production in *Escherichia coli*

Dear Dr Dias,

I am pleased to inform you that your manuscript has been formally accepted for publication in PLOS Computational Biology. Your manuscript is now with our production department and you will be notified of the publication date in due course.

With kind regards,

Alice Ellingham
